# Effectiveness of physical rehabilitation interventions in critically ill patients—A protocol for an overview of systematic reviews

**Ruvistay Gutierrez-Arias**[1,2]*, **Peter Nydahl**[3], **Dawid Pieper**[4,5], **Felipe González-Seguel**[6], **Yorschua Jalil**[7,8], **Maria-Jose Oliveros**[9,10], **Rodrigo Torres-Castro**[11], **Pamela Seron**[9,10]

**1** Servicio de Medicina Física y Rehabilitación, Unidad de Kinesiología, Instituto Nacional del Tórax, Santiago, Chile, **2** Exercise and Rehabilitation Sciences Laboratory, Faculty of Rehabilitation Sciences, School of Physical Therapy, Universidad Andres Bello, Santiago, Chile, **3** Department of Nursing Research, University Hospital of Schleswig-Holstein, Kiel, Germany, **4** Faculty of Health Sciences Brandenburg, Brandenburg Medical School (Theodor Fontane), Institute for Health Services and Health Systems Research, Rüdersdorf, Germany, **5** Center for Health Services Research, Brandenburg Medical School (Theodor Fontane), Rüdersdorf, Germany, **6** Carrera de Kinesiología, Facultad de Medicina, Clínica Alemana Universidad del Desarrollo, Santiago, Chile, **7** Departamento de Medicina Intensiva, Facultad de Medicina, Pontificia Universidad Católica de Chile, Santiago, Chile, **8** Departamento de Ciencias de la Salud, Carrera de Kinesiología, Facultad de Medicina, Pontificia Universidad Católica de Chile, Santiago, Chile, **9** Departamento de Ciencias de la Rehabilitación, Facultad de Medicina, Universidad de La Frontera, Temuco, Chile, **10** Centro de Excelencia CIGES, Universidad de La Frontera, Temuco, Chile, **11** Department of Physical Therapy, University of Chile, Santiago, Chile

* ruvistay.gutierrez@gmail.com

## Abstract

### Introduction

Adult and pediatric patients admitted to intensive care units (ICUs) requiring invasive ventilatory support, sedation, and muscle blockade may present neuromusculoskeletal deterioration. Different physical rehabilitation interventions have been studied to evaluate their effectiveness in improving critically ill patients' outcomes. Given that many published systematic reviews (SRs) aims to determine the effectiveness of different types of physical rehabilitation interventions, it is necessary to group them systematically and assess the methodological quality of SRs to help clinicians make better evidence-based decisions. This overview of SRs (OoSRs) aims to map the existing evidence and to determine the effectiveness of physical rehabilitation interventions to improve neuromusculoskeletal function and other clinical outcomes in adult and pediatric critically ill patients.

### Methods

An OoSRs of randomized and non-randomized clinical trials involving critically ill adult and pediatric patients receiving physical rehabilitation intervention will be conducted. A sensitive search of MEDLINE (Ovid), Embase (Ovid), CINAHL (EBSCOhost), Cochrane Library, Epistemonikos, and other search resources will be conducted. Two independent reviewers

**Data Availability Statement:** No datasets were generated or analysed during the current study. All

relevant data from this study will be made available upon study completion.

**Funding:** The author(s) received no specific funding for this work.

**Competing interests:** The authors have declared that no competing interests exist.

will conduct study selection, data extraction, and methodological quality assessment. Discrepancies will be resolved by consensus or a third reviewer. The degree of overlap of studies will be calculated using the corrected covered area. The methodological quality of the SRs will be measured using the AMSTAR-2 tool. The GRADE framework will report the certainty of evidence by selecting the "best" SR for each physical rehabilitation intervention and outcome.

## Discussion

The findings of this overview are expected to determine the effectiveness and safety of physical rehabilitation interventions to improve neuromusculoskeletal function in adult and pediatric critically ill patients based on a wide selection of the best available evidence and to determine the knowledge gaps in this topic by mapping and assessing the methodological quality of published SRs.

## Registration number

CRD42023389672.

## Introduction

Adults and pediatric patients admitted to intensive care units (ICUs) requiring invasive ventilatory support, sedation, and muscle blockade will develop neuromuscular, cognitive, and respiratory complications [1–5]. Neuromusculoskeletal deterioration may include loss of muscle mass and strength of limbs and respiratory muscles, altered nerve conduction, decreased range of motion associated with joint or postural alterations, and reduced bone mineral density, among others [6–9]. These complications can lead to poor clinical outcomes and functional prognosis. In the presence of impaired muscle function, increased time on mechanical ventilation [10, 11], increased total hospital and ICU stay [12], and even a higher mortality rate have been reported [13]. In addition, loss of muscle function has been associated with in-hospital cognitive impairment and persistent post-discharge complications resulting from ICU hospitalization [14].

Different interventions have been proposed and studied to evaluate their effectiveness and safety in improving critically ill patients' outcomes. Early mobilization is one of the most widely used physical rehabilitation interventions in both adult and pediatric critically ill patients used to improve peripheral and respiratory physical functioning, including progressive mobility, passive or active range of motion exercises, respiratory muscle training, cycle ergometry, and neuromuscular (NMES) or functional electrostimulation (FES) [15–17]. Early mobilization is safe [18], and some primary studies have reported its positive effect on physical functioning and delirium [19], and to reduce ICU and hospital length of stay [20–22].

Intervention systematic reviews (SRs) are considered to have the highest level of evidence to establish the effectiveness and safety of any intervention in different health conditions [23]. This type of secondary study is the basis for developing recommendations in clinical practice guidelines [24]. However, the number of SRs published recently has increased exponentially [25–27], and some SRs seek to answer similar research questions, sometimes exhibiting limited methodological quality. This has occurred in the critical care field for adults and pediatric

patients, especially considering non-pharmacological interventions such as physical rehabilitation. Many SRs report inconsistent results despite including similar primary studies.

Overviews of SRs (OoSRs), or umbrella reviews, have among their main objectives to collect information from multiple SRs of different health interventions for the same health condition and to map existing evidence on the topic to establish knowledge gaps [28]. Given that many published SRs aimed at determining the effectiveness and safety of physical rehabilitation interventions to improve functional and clinical outcomes of adult and pediatric critically ill patients and their heterogeneous treatment modalities, it is necessary to group them systematically and assess the methodological quality to help clinicians make better evidence-based decisions. Considering that the existing OoSRs on this topic need to be updated due to included SRs were published before October 2015 and mainly focused on adult patients, a significant number of SRs on this topic has grown significantly in recent years [26, 27]. Therefore, this OoSRs aim to map the current evidence and to determine the effectiveness and safety of physical rehabilitation interventions aimed at improving neuromusculoskeletal function and other clinical outcomes in adult and pediatric critically ill patients.

## Methods

An OoSRs will be conducted according to the methodology proposed by the Cochrane Handbook for Systematic Reviews of Interventions [28]. This protocol is reported according to Preferred Reporting Items for Systematic Review and Meta-Analysis Protocols (PRISMA-P) statement [29] and was registered in PROSPERO under the number CRD42023389672. The findings of this overview will be reported according to the Preferred Reporting Items for Overviews of Reviews (PRIOR) statement [30].

### Eligibility criteria

**Type of studies.** Intervention SRs, with or without meta-analysis, that have considered randomized (RCTs) or non-randomized clinical trials (non-RCTs) will be included. SRs that perform only network meta-analyses without including pairwise comparative analyses of interventions (conventional meta-analyses) will be excluded due to the indirect nature of the comparison of the interventions evaluated. Excluded will be non-systematic reviews, protocols, or abstracts.

There will be no limits in language or year of publication.

Considering that there are different definitions of SRs [31], for this overview, Intervention SRs will be defined as an evidence synthesis that aims to answer pre-defined research questions using explicit, reproducible methods to identify, critically appraise and combine results of primary research studies aimed at determining the effectiveness of any intervention on different health conditions [32].

**Type of participants.** SRs that consider adult or pediatric patients, with the majority (>50%) being on invasive or non-invasive mechanical ventilation at least once during the stay in ICU, will be included. The illness or health condition that led to the need for critical care shall not limit inclusion.

**Type of interventions.** SRs that consider rehabilitation interventions aimed at improving the neuromusculoskeletal and movement-related function of critically ill patients as defined by the International Classification of Functioning, Disability and Health (ICF) [33] will be included. These interventions should impact bone and joint, muscle, or movement functions [33]. They may have, but are not limited to, passive mobilization of limbs or another body segment [34, 35], exercises involving active patient participation [36], use of assistive devices such

as neuromuscular electrical stimulation (NMES) [15, 37], upper and lower extremity cycling or cycle ergometer [15, 34, 35], and respiratory muscle training [38], among others.

Physical rehabilitation interventions may be delivered as a single intervention or with other interventions (multimodal rehabilitation). However, interventions other than physical rehabilitation interventions should be delivered similarly in both groups (experimental and control) to isolate the effect of the physical interventions on the patient.

**Type of comparators.**  SRs that consider any intervention in the control groups of the primary studies will be included. These interventions may be usual care, placebo, sham, or other physical rehabilitation intervention.

**Types of outcomes.**  SRs that have addressed the effectiveness of physical rehabilitation interventions on at least one of the following outcomes will be included:

*Primary outcomes.*

- Mobility: Outcome that can be measured with any generic or specific scale to assess mobility in the ICU, such as Functional Status Score for the Intensive Care Unit (FSS-ICU) [39], ICU mobility scale (IMS) [40], The Chelsea Critical Care Physical Assessment Tool (CPAx) [41], or any other measure to assess mobility.

- Muscle strength: Outcome that can be measured using manual muscle testing, for example, MRC-SS [42], or using a device that allows the assessment of handgrip strength [43] or the pressures generated by the respiratory muscles [44], among others.

- Muscle mass: Outcome which can be measured by muscle circumference measurement, ultrasonography, dual-energy X-ray absorptiometry, and computed tomography scan [45], among others.

*Secondary outcomes.*

- Duration of mechanical ventilation: number of days patients remain on invasive ventilatory support.

- ICU length of stay: days between admission to the ICU and discharge to a less complex unit.

- Mortality: Due to any cause and which can be reported according to different follow-up points, for example, mortality in ICU, hospital, 90 days, 180 days, 360 days, the number of deaths due to a given cause.

- Incidence and duration of delirium: Outcome that can be measured with a scale such as the Confusion Assessment Method for the Intensive Care Unit (CAM-ICU) [46], among others.

- Unwanted safety events: Outcome that can be measured as the incidence of any unwanted safety events associated with delivering physical rehabilitation interventions reported by SRs.

## Search strategy

A systematic search with a sensitive approach will be conducted in different electronic databases and other search resources. MEDLINE (through Ovid), Embase (through Ovid), CINAHL (through EBSCOhost), Cochrane Library, and Epistemonikos will be searched using controlled language (i.e., MeSH, Emtree, and CINAHL Subject Headings) and key terms. In addition, the International Prospective Register of Systematic Reviews (PROSPERO), International Platform of Registered Systematic Review and Meta-analysis Protocols (INPLASY), and Open Science Framework (OSF) registries will be reviewed to identify complete SRs that the electronic database search may not have identified.

In addition, the references of the SRs included in this overview will be manually searched using the Citationchaser tool (forward and backward citation) [47], and experts in critically ill patient rehabilitation will be consulted to identify potential SRs that meet the eligibility criteria of this overview.

The search strategy for MEDLINE (Ovid) (Table 1) was constructed following the Peer Review of Electronic Search Strategies (PRESS) statement [48], which will be adapted for the other electronic databases and search resources. The Canadian Agency for Drugs and Technologies in Health (CADTH) filter will be used to identify studies with an SR design [49].

The search strategy will not use language or publication status restrictions.

## Study selection

Two reviewers will independently check records identified by the search strategy for compliance with the eligibility criteria. Irrelevant records will be excluded by reading the title and abstract and then determining the inclusion of SRs by reading the full text. Disagreements will be resolved by consensus or by the involvement of a third reviewer. The Rayyan® application will be used to improve the efficiency of this overview stage [50].

## Data extraction

Two reviewers will independently extract data from the SRs included in this overview. A bespoke extraction form created for this study will be used. As an iterative approach, we will pilot the extraction form following a data extraction process from 5 SRs, then adapted according to the reviewers' feedback in the piloting. This form will be used to extract data describing bibliometric characteristics, general characteristics of the SRs, reported outcome data, quality or risk of bias of the primary studies included, and certainty of evidence (Table 2). In addition, the methodological quality of the included SRs can be rated in this form. Potential disagreements will be resolved by consensus or by the involvement of a third reviewer.

## Methodological appraisal

Two reviewers will independently assess the methodological quality of the SRs included in this overview using "A MeaSurement Tool to Assess systematic Reviews 2" (AMSTAR 2) [51]. Disagreements will be resolved by consensus or by the involvement of a third reviewer.

This tool includes 16 items and considers seven as critical:

1. Protocol registered before the commencement of the review;

2. Adequacy of the literature search;

3. Justification for excluding individual studies;

4. Risk of bias from individual studies being included in the review;

5. Appropriateness of meta-analytical methods;

6. Consideration of risk of bias when interpreting the results of the review;

7. Assessment of presence and likely impact of publication bias.

SRs will be classified according to the overall confidence in their results as High, Moderate, Low, and Critically Low, according to the following criteria:

- High: No or one non-critical weakness. The SR provides an accurate and comprehensive summary of the results of the available studies that address the question of interest.

**Table 1. Search strategy for MEDLINE (Ovid).**

| N° | Search term |
|----|-------------|
| 1 | Exercise/ |
| 2 | exp Exercise Therapy/ |
| 3 | exp Rehabilitation/ |
| 4 | exp Physical Therapy Modalities/ |
| 5 | Occupational Therapy/ |
| 6 | "Physical Therapy (Specialty)"/ |
| 7 | "activities of daily living"/ |
| 8 | early ambulation/ |
| 9 | recovery of function/ or movement/ or locomotion/ or walking/ or motor activity/ or exercise movement techniques/ |
| 10 | exercis$.tw. |
| 11 | (mobilizat$ or mobilisat$ or mobility).tw. |
| 12 | (therap$ adj3 (physical or exercise or occupation$)).tw. |
| 13 | ((bed or daily living) adj3 activit$).tw. |
| 14 | (training or pregait or pre-gait or walk$ or adl or physiotherap$ or ambulation).tw. |
| 15 | ((cycle or bicycle) adj2 ergomet$).tw. |
| 16 | or/1-15 |
| 17 | exp Electric Stimulation Therapy/ |
| 18 | ((neuro$ or musc$ or electr$ or nerve) adj3 stim$).tw. |
| 19 | electrotherap$.tw. |
| 20 | myostim$.tw. |
| 21 | electrostim$.tw. |
| 22 | electroneurostim$.tw. |
| 23 | neurostim$.tw. |
| 24 | (EMS or FES or TENS or NMES).tw. |
| 25 | or/17-24 |
| 26 | exp Breathing Exercises/ |
| 27 | ((respir$ or inspirat$ or expiratory or ventilatory or chest) adj4 (training or exercise$ or endurance)).tw. |
| 28 | (voluntar$ adj3 isocapn$ adj3 hyperpnoe$).tw. |
| 29 | (threshold adj3 (load or device$)).mp. |
| 30 | resistive breathing.mp. |
| 31 | (resist$ adj3 load$).tw. |
| 32 | or/26-31 |
| 33 | 16 or 25 or 32 |
| 34 | Critical Illness/ |
| 35 | exp Intensive Care Units/ |
| 36 | exp Critical Care/ |
| 37 | (intensive care or intensive-care or critical care or critical-care).tw. |
| 38 | (icu or icuaw or icu-aw).tw. |
| 39 | (critical$ adj3 (ill$ or care$)).tw. |
| 40 | ((intubat$ or ventilat$) adj5 patient$).tw. |
| 41 | or/34-40 |
| 42 | 33 and 41 |
| 43 | (systematic review or meta-analysis).pt. |
| 44 | meta-analysis/ or systematic review/ or systematic reviews as topic/ or meta-analysis as topic/ or "meta analysis (topic)"/ or "systematic review (topic)"/ or exp technology assessment, biomedical/ or network meta-analysis/ |
| 45 | ((systematic$ adj3 (review$ or overview$)) or (methodologic$ adj3 (review$ or overview$))).ti,ab,kf. |
| 46 | ((quantitative adj3 (review$ or overview$ or synthes$)) or (research adj3 (integrati$ or overview$))).ti,ab,kf. |

*(Continued)*

**Table 1.** (Continued)

| N° | Search term |
|---|---|
| 47 | ((integrative adj3 (review$ or overview$)) or (collaborative adj3 (review$ or overview$)) or (pool$ adj3 analy$)).ti,ab,kf. |
| 48 | (data synthes$ or data extraction$ or data abstraction$).ti,ab,kf. |
| 49 | (handsearch$ or hand search$).ti,ab,kf. |
| 50 | (mantel haenszel or peto or der simonian or dersimonian or fixed effect$ or latin square$).ti,ab,kf. |
| 51 | (met analy$ or metanaly$ or technology assessment$ or HTA or HTAs or technology overview$ or technology appraisal$).ti,ab,kf. |
| 52 | (meta regression$ or metaregression$).ti,ab,kf. |
| 53 | (meta-analy$ or metaanaly$ or systematic review$ or biomedical technology assessment$ or bio-medical technology assessment$).mp,hw. |
| 54 | (medline or cochrane or pubmed or medlars or embase or cinahl).ti,ab,hw. |
| 55 | (cochrane or (health adj2 technology assessment) or evidence report).jw. |
| 56 | (comparative adj3 (efficacy or effectiveness)).ti,ab,kf. |
| 57 | (outcomes research or relative effectiveness).ti,ab,kf. |
| 58 | ((indirect or indirect treatment or mixed-treatment or bayesian) adj3 comparison$).ti,ab,kf. |
| 59 | (meta-analysis or systematic review).mp. |
| 60 | (multi$ adj3 treatment adj3 comparison$).ti,ab,kf. |
| 61 | (mixed adj3 treatment adj3 (meta-analy$ or metaanaly$)).ti,ab,kf. |
| 62 | umbrella review$.ti,ab,kf. |
| 63 | (multi$ adj2 paramet$ adj2 evidence adj2 synthesis).ti,ab,kf. |
| 64 | (multiparamet$ adj2 evidence adj2 synthesis).ti,ab,kf. |
| 65 | (multi-paramet$ adj2 evidence adj2 synthesis).ti,ab,kf. |
| 66 | or/43-65 |
| 67 | 42 and 66 |

- Moderate: More than one non-critical weakness. The SR has more than one weakness but no critical flaws. It may provide an accurate summary of the results of the available studies that were included in the review.

- Low: One critical flaw with or without non-critical weaknesses. The SR has a critical flaw and may not provide an accurate and comprehensive summary of the available studies that address the question of interest.

- Critically low. More than one critical flaw with or without non-critical weaknesses. The SR has more than one critical flaw and should not be relied on to provide and accurate and comprehensive summary of the available studies.

## Certainty of the evidence

The "Grades of Recommendation, Assessment, Development, and Evaluation" (GRADE) framework will be used to report the certainty of evidence [52]. Study limitations, inconsistency of results, Indirectness of evidence, imprecision, and reporting bias are assessed by the GRADE approach [53]. According to this framework, the certainty of the evidence can be qualified as:

- High quality: We are very confident that the true effect lies close to that of the estimate of the effect.

- Moderate quality: We are moderately confident in the effect estimate. The true effect is likely to be close to the estimate of the effect, but there is a possibility that it is substantially different.

**Table 2. Data extraction.**

| Domain | Data to extract |
|---|---|
| Bibliometric characteristics | a) First author |
| | b) Year of publication |
| | c) Journal |
| | d) Impact factor |
| General characteristics of the SRs | a) Number of included studies |
| | b) Meta-Analysis (yes/no) |
| | c) Population description |
| | d) Number of included patients (total, intervention, control) |
| | e) Age of included patients (pediatric, adult, geriatric, mixed) |
| | f) Physical rehabilitation interventions considered by the SR |
| | g) Electronic databases and other search resources considered by the SR |
| | h) Search timeframe and languages |
| | i) Study designs included by the SR |
| | j) Profession performing intervention |
| Reported outcome data | a) Outcomes initially considered by SRs |
| | b) Outcomes reported by SRs |
| | c) Scales, scores, questionnaires, and biophysical instruments used to assess different outcomes |
| | d) Results data for each outcome reported |
| | e) Intensity (in bed vs. out of bed) |
| | f) Duration (in total, per day, or <30min vs. $\geq$ 30 min per day) |
| | g) Frequency (daily 7/7, Mo-Fr (5/7), Mo-Sa 6/7) |
| | h) Initiation (1-3rd day, 4-7th day, $\geq$8days) |
| Quality or risk of bias of the primary studies | a) Instrument for assessing the methodological quality or risk of bias of included primary studies |
| | b) Results of the assessment of the methodological quality or risk of bias of the included studies |
| Certainty of evidence | a) Instruments or framework used to assess the certainty of the evidence |
| | b) Results of the assessment of the certainty of the evidence |

- Low quality: Our confidence in the effect estimate is limited. The true effect may be substantially different from the estimate of the effect.

- Very low quality: We have very little confidence in the effect estimate. The true effect is likely to be substantially different from the estimate of effect.

Where the SR chosen to report the effectiveness of physical rehabilitation interventions on the different outcomes has assessed the certainty of the evidence according to the GRADE framework, this information will be extracted and presented. Otherwise, the necessary information will be extracted from the SRs to perform their assessment.

## Data analysis

The unit of analysis of this overview will be the SR. Therefore, the primary studies included by each SR will not be accessed in case of missing data.

Due to the possible existence of redundant SRs, i.e., seeking to answer the same research question [54], and the likelihood of finding different degrees of primary study overlap between these SRs [55], strategies will be applied to visualize [56], calculate [55], and managing the

overlap [57, 58]. Different criteria will be considered to choose the "best" SR for reporting the effectiveness of various physical rehabilitation interventions on the outcomes considered by this overview.

First, a matrix will be created that cross-references the SRs included in this overview with the primary studies included by these SRs. This will be done at the SR and outcome level. In addition, from these matrices, the corrected covered area (CCA) [55] will be calculated without considering any structural missing data and considering the chronological structural missing data and by primary study design. The ccaR package (https://github.com/thdiakon/ccaR) will be used [59].

To select the "best" SR for reporting the effectiveness of each physical rehabilitation intervention and outcome, the following decision algorithm will be followed:

- Comprehensive summary and methodological quality of the SRs: the SR contains the highest number of primary studies in the group of SRs rated with high or moderate overall confidence in their results according to AMSTAR 2 will be chosen. Priority will be given to SRs with a high-quality rating according to AMSTAR 2.

- Primary study design: if there are two or more SRs with an equal number of included primary studies and methodological quality, the one including only RCTs will be chosen over those including RCTs + non-RCTs or non-RCTs.

The different analyses proposed will be carried out, including all SRs, and by subgroup according to the age of the population included (adult vs. pediatric).

### Evidence synthesis

The SR selection process will be reported in narrative form with a PRISMA-type flow chart [60].

The characteristics and results of all SRs will be presented in narrative form using tables and figures. This will be done for all SRs and separated by population, intervention, and outcomes.

The crossover matrix of the SRs and primary studies included will be reported. In addition, heat map graphics will be presented to inform the degree of overlap of prior studies at the SR and outcome level.

The effectiveness of the different physical rehabilitation interventions for each outcome of interest will be presented in narrative form accompanied by a summary of findings (SoF) tables [61, 62].

## Discussion

The results of this overview are expected to determine the effectiveness and safety of physical rehabilitation interventions to improve neuromusculoskeletal function in adult and pediatric critically ill patients based on the selection of the best available evidence and to determine the knowledge gaps in this topic by mapping and assessing the methodological quality of published SRs.

The need for new SRs answering the same research question should be assessed. Review of SR protocol records [63] should be mandatory to reduce redundancy and research waste.

This OoSRs has advantages worth mentioning. First, SRs of adult and pediatric critically ill patients will be included, widening the scope of knowledge on early mobilization in the ICU. Considering that most studies on early mobilization have been reported for adult patients, this OoSRs adds new evidence for the growing knowledge on critically ill children. Second, this OoSRs will include the SRs from the last five years, which have been more than those from

previous years. In addition, evidence comprising patients with or without COVID-19 during the pandemic period will also be included. Finally, this work will not be limited to one specific physical intervention since it will include SRs with all types of physical rehabilitation intervention used to improve the functional outcomes of critically ill patients.

This OoSRs has unique challenges that could become potential limitations. One of them is that our unit of analysis will be the SRs, so there may be missing information when it comes to extracting the information needed to meet our objectivesdata. In addition, physical rehabilitation interventions are often applied in conjunction with other rehabilitation interventions, which could lead to high heterogeneity of SRs, limiting the analysis. And finally, selecting the "best" SR for each physical rehabilitation intervention and outcome measurement is a challenge, as it is possible to lose some information (fewer primary studies) to optimize methodological quality or vice versa. This methodological issue has been studied without clear recommendations to solve this challenge [64].

## Supporting information

**S1 Checklist. PRISMA-P 2015 checklist.**
(DOCX)

## Acknowledgments

### Ethics and dissemination

As a synthesis of evidence, this study does not involve the participation of people whose rights may be violated. However, this overview will be developed rigorously and systematically to achieve valid and reliable results.

The findings of overview SRs will be presented at conferences and published in a peer-reviewed journal related to rehabilitation or critical care.

## Author Contributions

**Conceptualization:** Ruvistay Gutierrez-Arias, Peter Nydahl, Dawid Pieper, Felipe González-Seguel, Yorschua Jalil, Maria-Jose Oliveros, Rodrigo Torres-Castro, Pamela Seron.

**Methodology:** Ruvistay Gutierrez-Arias, Peter Nydahl, Dawid Pieper, Pamela Seron.

**Project administration:** Ruvistay Gutierrez-Arias.

**Supervision:** Ruvistay Gutierrez-Arias.

**Writing – original draft:** Ruvistay Gutierrez-Arias.

**Writing – review & editing:** Peter Nydahl, Dawid Pieper, Felipe González-Seguel, Yorschua Jalil, Maria-Jose Oliveros, Rodrigo Torres-Castro, Pamela Seron.

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
