## [Decision Letter · Decision Letter 0]

8 Mar 2023

PONE-D-23-01741Effectiveness of physical rehabilitation interventions to improve neuromusculoskeletal function in critically ill patients. Protocol for an overview of systematic reviewsPLOS ONE

Dear Dr. Gutierrez-Arias,

Thank you for submitting your manuscript to PLOS ONE. After careful consideration, we feel that it has merit but does not fully meet PLOS ONE’s publication criteria as it currently stands. Therefore, we invite you to submit a revised version of the manuscript that addresses the points raised during the review process.

We look forward to receiving your revised manuscript.

Kind regards,

Jan René Nkeck, M.D., M.Sc

Academic Editor

PLOS ONE

Journal Requirements:

Reviewers' comments:

Reviewer's Responses to Questions

**Comments to the Author**

1. Does the manuscript provide a valid rationale for the proposed study, with clearly identified and justified research questions?

Reviewer #1: Yes

Reviewer #2: Yes

2. Is the protocol technically sound and planned in a manner that will lead to a meaningful outcome and allow testing the stated hypotheses?

Reviewer #1: Partly

Reviewer #2: Partly

3. Is the methodology feasible and described in sufficient detail to allow the work to be replicable?

Reviewer #1: Yes

Reviewer #2: Yes

4. Have the authors described where all data underlying the findings will be made available when the study is complete?

Reviewer #1: Yes

Reviewer #2: Yes

5. Is the manuscript presented in an intelligible fashion and written in standard English?

Reviewer #1: Yes

Reviewer #2: Yes

6. Review Comments to the Author

You may also provide optional suggestions and comments to authors that they might find helpful in planning their study.

Reviewer #1: Introduction

I would like more context around the various populations and interventions that could be included here as the eligibility seems wide so coverage should be equally wide

Methods

Are references 24 and 25 the best reference to use doesn’t it refer to a standard systematic review of interventions not an overview review? Doesn’t reference 26 cover what you want to do – how does reference 24 and 25 supplement 26?

Can you justify why you are excluding some SRs with meta-analysis and add a reference

For eligibility around the type of intervention can you explain how you will deal with complex interventions that have multiple components where one is those you name but others are included

Re AMSTAR May be worth contacting Lu et al 2020 for provisional findings from their work or obtain a reference to support choices made

Can you identify if GRADE is designed to deal with a collection of systematic reviews rather than collection of interventions? Does the reader need to understand via a supplementary file what you will assess and how you will allocate points for upgrading and downgrading. I ma not clear how you will assess study inconsistency of results, indirectness of evidence, imprecision and reporting bias e.g., ROBIS is better than AMSTAR at reporting bias

Not convinced at how the analysis will account for different interventions, conditions, reviews with adults vs children may be a supplementary file could help this

Discussion is limited and I think you can expand based on the fact you have a very large and open consideration to all interventions and many population groups.

Reviewer #2: There are some grammatical issues with the topic of the topic. After the phrase critically ill patients, put a column and not a full stop. Start the next part with an article 'a' to read, 'A Protocol for an overview....'

The sentence on line 72 which extends to line 74 should be reformulated to follow grammatical rules.

You intend to review studies which look at both children and adults. Their musculo-skeletal adaptation is not the same will that not play on the outcome of this study? Is it not better to use just studies that include children or those involving just adults

7. PLOS authors have the option to publish the peer review history of their article (what does this mean?). If published, this will include your full peer review and any attached files.

Reviewer #1: No

Reviewer #2: No

---

## [Author Response · Author response to Decision Letter 0]

14 Mar 2023

The responses to reviewers and editors were uploaded in a document entitled "Response to reviewers".

---

## [Decision Letter · Decision Letter 1]

30 Mar 2023

Effectiveness of physical rehabilitation interventions in critically ill patients - A protocol for an overview of systematic reviews

PONE-D-23-01741R1

Dear Dr. Gutierrez-Arias,

We’re pleased to inform you that your manuscript has been judged scientifically suitable for publication and will be formally accepted for publication once it meets all outstanding technical requirements.

Kind regards,

Jan René Nkeck, M.D., M.Sc

Academic Editor

PLOS ONE

Additional Editor Comments (optional):

Reviewers' comments:

Reviewer's Responses to Questions

**Comments to the Author**

1. Does the manuscript provide a valid rationale for the proposed study, with clearly identified and justified research questions?

Reviewer #1: Yes

Reviewer #2: Yes

2. Is the protocol technically sound and planned in a manner that will lead to a meaningful outcome and allow testing the stated hypotheses?

Reviewer #1: Yes

Reviewer #2: Yes

3. Is the methodology feasible and described in sufficient detail to allow the work to be replicable?

Reviewer #1: Yes

Reviewer #2: Yes

4. Have the authors described where all data underlying the findings will be made available when the study is complete?

Reviewer #1: Yes

Reviewer #2: Yes

5. Is the manuscript presented in an intelligible fashion and written in standard English?

Reviewer #1: Yes

Reviewer #2: Yes

6. Review Comments to the Author

You may also provide optional suggestions and comments to authors that they might find helpful in planning their study.

Reviewer #1: Thanks for attending to changes suggested. I believe interesting results will come through. Best wishes for your work.

Reviewer #2: The comments raised during review have been addressed except the topic which should read

Effectiveness of physical rehabilitation interventions in critically ill patients: A protocol

for an overview of systematic reviews

7. PLOS authors have the option to publish the peer review history of their article (what does this mean?). If published, this will include your full peer review and any attached files.

Reviewer #1: **Yes: **A Soundy

Reviewer #2: **Yes: **Leonard Ngarka

---

## [Editor Report · Acceptance letter]

4 Apr 2023

PONE-D-23-01741R1 

Effectiveness of physical rehabilitation interventions in critically ill patients − A protocol for an overview of systematic reviews 

Dear Dr. Gutierrez-Arias:

I'm pleased to inform you that your manuscript has been deemed suitable for publication in PLOS ONE. Congratulations! Your manuscript is now with our production department. 

Kind regards, 

on behalf of

Dr. Jan René Nkeck 

Academic Editor

PLOS ONE